# Green Financing Efficiency and Influencing Factors of Chinese Listed Construction Companies against the Background of Carbon Neutralization: A Study Based on Three-Stage DEA and System GMM

**Yaguai Yu [1,2], Yina Yan [1], Panyi Shen [1,\*], Yuting Li [1] and Taohan Ni [3]**

1    Business School, Ningbo University, Ningbo 315211, China
2    Donghai Academy, Ningbo University, Ningbo 315211, China
3    Business School, University of Nottingham, Ningbo 315199, China
\*    Correspondence: shenpanyi1998@126.com

**Abstract:** This paper combines the green industrial strategy and green financial policies for the construction industry implemented in China in the context of carbon neutrality. A total of 67 listed companies in the construction industry from 2017 to 2020 were taken as the research sample, the green financing efficiency was measured, and its influencing factors were identified based on the three-stage DEA and systematic GMM method. The findings show that the green financing efficiency of listed companies in the construction industry is not high overall, although it is increasing. There are obvious differences in subsectors, among which, the efficiency of architectural design and service industries is relatively high. Overall, the financial environment, and the interaction between the government and the financial market, significantly and positively influence the green financing efficiency. In addition, the macroeconomic environment and the government–enterprise relationship has a complex impact on the green financing efficiency. The ownership concentration and having corporate executives with a financial background have a significant positive impact on the green financing efficiency, and the enterprise size, the debt maturity structure, and the R&D and innovation capability have a significant negative impact. The findings of this paper have implications for the improvement of the policy system that supports green development in the construction industry, and provide guidance for the strategic adjustment of the construction industry itself.

**Keywords:** green financing efficiency; green financing efficiency impact factors; construction industry; industry differences

## 1. Introduction

In the context of carbon neutrality, the construction industry, as an important pillar industry of the national economy, faces the constraints of external environmental regulations. In addition, it must stop using the traditional inefficient development mode; hence, a green and low-carbon transformation is imminent. In the green and low-carbon transformation of the construction industry, due to the positive externalities of green buildings, combined with the asymmetry of market information, the lack of a credit system, and poor internal management, the market-oriented financing of green projects faces more obstacles that hinder the process of green and low-carbon development.

Although China has been vigorously promoting the construction of the green financial system since 2015, and including green buildings in the support system, under the constraints of technology, system, and other aspects, green finance is caught in the development paradox of "sustainable operation or environmental sustainability" (Li Xiaoxi, 2017) [1], and the actual support for the construction industry remains to be examined. Therefore, in a scenario of limited funds, it is important to accurately assess the green financing efficiency

of the construction industry and effectively identify the influencing factors to help the industry develop and improve its green financing efficiency.

The contributions of this study are as followings. First, an evaluation index system of green financing efficiency under carbon neutrality was constructed, with a focus on the green financing efficiency of listed companies in the construction industry, in combination with the three-stage DEA method to support understanding of the current situation of green financing efficiency in the construction industry. Second, the influencing factors of the green financing efficiency of listed companies in the construction industry were analyzed in terms of both internal and external aspects, which helps the government and construction enterprises to prescribe remedies to improve the green financing efficiency of listed companies in the construction industry. Third, the green financing efficiency of listed companies in the construction industry under the green finance system was analyzed, which helps to enrich the relevant research on green finance and provides a reference for the subsequent implementation of green finance policies.

The rest of this paper is structured as follows: Part II is a literature review; Part III presents the research methodology and index selection; Part IV presents a green financing efficiency measure for listed companies in the construction industry; Part V presents an analysis of the internal influencing factors of green financing efficiency for listed companies in the construction industry; Part VI presents an extended analysis; and Part VII presents a conclusion.

## 2. Literature Review

Most of the early studies on green financing efficiency were affiliated with green finance, and although more literature did not address the definition, in actual research, scholars generally defined the act of relying on green financial instruments to obtain funds as green financing and regarded the effectiveness of this act as green financing efficiency. Guo Chaoxian et al. (2015) [2] regard the efficiency of the use of funds incorporated by environmental industries through green financial instruments as green financing efficiency. In looking at the development of green finance, Ma Jun (2016) [3] refers to the act of using green bond markets, green stock indices, and related products as green financing, and the effectiveness of green financing as green financing efficiency. Lu Zhengwei and Fang Qi (2018) [4] take the effectiveness of green credit as the green financing efficiency of the banking sector by examining the practice of green credit. Liu, Xiliang and Wenyang (2019) [5] include the efficiency of environmental responsibility fulfillment in the category of green financing efficiency based on the previous studies.

With the in-depth development of green finance, the assessment of green financing efficiency has received more and more attention, and relevant research directions have gradually shifted to the measurement of green financing efficiency, and focused on exploring green financing efficiency from both input and output dimensions. Earlier, Motoko Aizawa and Chaofei Yang (2010) [6], and Murillo Campello and John R. Graham (2013) [7], from the perspective of quantitative analysis, adopted the ratio of the number of commercial banks joining the Equator Principles and the volume of green credit indicators, such as the volume of green credit issuance, to measure the country's green financing efficiency. Based on the previous work, domestic scholars have gradually constructed a green financing efficiency input–output index system to conduct relevant research. Zhang Lili et al. (2018) [8] used the entropy value method and DEA–Malmquist index to measure the green financing efficiency at the national, inter-provincial, and regional levels, using the funds incorporated in green credit, green securities, green insurance, green investment, and carbon finance as input indicators, and the economic contribution and social contribution of enterprises as output indicators. You, Soldier and Yang, Fang (2019) [9] used a non-radial directional distance function considering non-desired outputs, a composite index containing capital inputs, labor inputs, and regional energy inputs as input indicators, value-added of the real economy as the desired output indicator, and $CO_2$ emissions as a non-desired output

indicator to measure the green financing efficiency when financial services are provided to the real economy.

When green financing efficiency is measured, the influencing factors of green financing efficiency are also considered to be a key issue for discussion, and these factors are divided into two main categories. The first category is external influencing factors, which mainly revolve around the macroeconomic environment, financial environment, and government–enterprise relationship. Zhou Yujing and Luo Yunxuan (2017) [10] constructed a multiple regression model empirically proposing that green financing efficiency is lower in regions with a poor macroeconomic environment, using A-share listed companies in heavy pollution industries as a sample. Shiyi Chen (2021) [11] used the multiplicative difference method to examine the effect of financing constraints on the policy effect and proposed that a good financial environment can improve the green financing efficiency of enterprises. Qiao-Xin Xie and Yu Zhang (2021) [12] constructed a quasi-natural experiment with the implementation of the Green Credit Guidelines in 2012, and found that the supply of government subsidies under a good government–enterprise relationship is beneficial for firms to improve their green financing efficiency. The second category is internal influencing factors, which focus on firm characteristics, including enterprise size, debt maturity structure, ownership concentration, R&D, and innovation capacity. Jin Yi et al. (2021) [13], based on data of listed energy-saving and environmental protection firms in China from 2010 to 2019, and after empirical analysis using the Tobit model, suggested that green credit provided by banks in China's financial markets dominates green financing sources, and the size of the enterprise will have an impact on the efficiency of green financing of the enterprise. Wang Kangsi et al. (2019) [14] used 260 green firms in A-shares from 2010 to 2015 as a sample, and, in an empirical investigation of the mechanism of the impact of green finance development on green firm investment, they proposed that improving the debt maturity structure of green firms can improve their financing efficiency. Qian Wang and Xinda Li (2021) [15] analyzed mixed cross-sectional data based on 40 listed companies that issued green bonds from 2016 to 2021 and found that higher ownership concentration has a more negative impact on the actual internal management of the company. Pengfei Ge et al. (2018) [16] used cross-country panel data of "One Belt, One Road" to construct a benchmark model and empirically found that financing constraints can be alleviated and green financing efficiency can be improved through innovative channels.

From the above literature, the existing green financing efficiency index system is still not comprehensive in considering both green and financing elements, and fails to reflect the carbon reduction target. Furthermore, most previous studies have applied traditional DEA to examine green financing efficiency, ignoring the influence of environmental factors and random errors on the efficiency level. The analysis of the influencing factors is also not comprehensive; in particular, in the analysis of internal influencing factors, little attention has been paid to the characteristics of executives. As a result, in this study, the three-stage DEA and systematic GMM method were combined to construct a green financing efficiency index system that reflects green and carbon reduction, and more comprehensive influencing factors, including executive characteristics, were selected to measure green financing efficiency and explore the influencing factors.

Compared with previous studies, the innovations of this study are: firstly, the evaluation indexes of green financing efficiency were designed with "green" as the key analytical focus, and carbon dioxide and major pollutants were included in the index system in conjunction with the carbon peaking and carbon neutral strategies. Secondly, for the specific evaluation of green financing efficiency, an analysis system including the three-stage DEA and systematic GMM method was established. Three-stage DEA eliminates the interference of external influences and random errors to more accurately evaluate the green financing efficiency of listed companies in the construction industry, and the systematic GMM method can effectively alleviate the endogeneity problems caused by the first-order lagged terms of the explanatory variables and panel effect correlation. Third, the characteristics of listed companies in the construction industry were combined and executive characteristics were introduced as internal influencing

factors of their green financing efficiency. There are large differences in the background levels of executives of listed companies in the construction industry, and managerial traits influence their strategic choices and, in turn, affect corporate behavior.

## 3. Research Methodology and Indicator Selection

### 3.1. Research Methodology

In the context of carbon neutrality, the green financing efficiency of listed companies in the construction industry measures the ability to integrate green funds at the lowest cost and risk, and use the integrated green funds to create the greatest economic and environmental benefits, which is in line with the characteristics of data envelopment analysis (DEA). While three-stage DEA retains the advantages of DEA, it also has the advantages of SFA, which separates the interference of external influences and random errors, can be used to estimate the efficiency of different production units more accurately, and more closely matches the real operating conditions of enterprises (H.O. Fried et al., 2002) [17]. Therefore, this study adopted the three-stage DEA method to measure the green financing efficiency of listed companies in the construction industry.

The specific ideas for the application of the three-stage DEA are as follows.

In the first stage, the initial efficiency values are measured using the original input–output data. The original input and output data are substituted into the traditional DEA model to measure the efficiency of each decision unit. The DEA model can be further divided into CCR and BCC models. Under the CCR model, listed companies in the construction industry can expand their outputs by increasing inputs in equal proportion in the green financing-production process, i.e., changes in the scale of inputs will not affect their efficiency, but in practice it is difficult to operate practically due to various factors. The BCC model is an improvement of the CCR model, which relaxes the assumption of constant returns to scale under the CCR model and measures the combined technical efficiency (TE) of each subject based on the assumption of variable returns to scale, and decomposes it into scale efficiency (SE) and pure technical efficiency (PTE) (R.D. Banker et al., 1984) [18]. In the analysis process of this study, the input-oriented BCC-DEA model was used, and the pairwise model of the input-oriented BCC-DEA model is shown in Equation (1).

$$\min \theta - \varepsilon \left( e^T S^- + e^T S^+ \right) \tag{1}$$

$$\begin{cases} \sum\limits_{j=1}^{n} X_j \lambda_j + S^- = \theta X_0 \\ \sum\limits_{j=1}^{n} Y_j \lambda_j - S^+ = Y_0 \end{cases}$$

$\lambda_j \geq 0$, $S^- \geq 0$, $S^+ \geq 0$, j = 1, 2. . . . , *n* denotes the decision unit; X, Y denotes the input and output vectors. $S^+$ and $S^-$ are slack variables in the pairwise model, and $\varepsilon$ denotes the non-Archimedean infinitesimal quantity. Based on the assumption of variable returns to scale to examine the efficiency of the decision unit, the measured $\theta$ or TE, can be decomposed into the product of SE and PTE, as shown in Equation (2).

$$TE = SE * PTE \tag{2}$$

If $\theta = 1$, $S^+ = S^- = 0$, then the decision cell DEA is valid; if $\theta = 1$, $S^+ \neq 0$, or $S^- \neq 0$ then the decision unit weak DEA is valid; if $\theta < 1$, then the decision unit non-DEA is valid.

In the second stage, SFA is used to exclude the effects of external influences and random errors on the inputs, and finally, the input redundancy of the decision unit caused by management inefficiency only is obtained. First, the slack values of each input variable obtained in the first stage are calculated, and the input slack variables are used as the explanatory variables to exclude the effects of external influences and random errors on

the efficiency of green financing using SFA regression in the form of the function shown in Equation (3).

$$S_{ij} = f^i\left(z_j; \beta^i\right) + u_{ij} + v_{ij} \qquad (3)$$

$S_{ij}$ is the redundant variable for the jth input variable of the ith decision unit (i.e., input redundancy). $f^i\left(z_j; \beta^i\right)$ denotes the effect of environmental variables on input redundancy $S_{ij}$. $z_j = (z_{1j}, z_{2j}, ..., ... z_{pj})$, are the p observable environmental variables; the parameter vector $\beta^i$ represents the unknown parameters to be estimated. $u_{ij} + v_{ij}$ is the combined error term, where $u_{ij}$ reflects the management inefficiency, distribution $u_{ij} \sim N^+\left(u_i, \sigma_{ui}^2\right)$; and $v_{ij}$ denotes the random error, distribution $v_{ij} \sim N\left(0, \sigma_{vi}^2\right)$. Let $\gamma = \sigma_{ui}/(\sigma_{ui} + \sigma_{vi})$, representing the proportion of the variance of management inefficiency to the total variance; when the value of $\gamma$ tends to 1, it indicates that the management inefficiency factor is the main influence, and when the value of $\gamma$ tends to 0, it indicates that the influence of the random error factor is too large, at which time $u_{ij}$ can be eliminated.

Then, the regression results of Equation (3) are used to adjust the input variables so that all decision units are adjusted to the same environmental conditions, while excluding the influence of random error interference, so as to measure the actual input values excluding the influence of external influencing factors and the influence of random error. The adjustment method is shown in Equation (4).

$$X_{ij}^A = X_{ij} + \left[\max f\left(z_j; \hat{\beta}^i\right) - f\left(z_j; \hat{\beta}^i\right)\right] + \left[\max(v_{ij}) - v_{ij}\right] \qquad (4)$$

$X_{ij}^A$ and $X_{ij}$ are adjusted input values and pre-adjusted input values, respectively, $\hat{\beta}^i$ is the coefficient to be estimated for the environmental variables, $\max f\left(z_j; \hat{\beta}^i\right) - f\left(z_j; \hat{\beta}^i\right)$ denotes adjusting all decision units to homogeneous external environmental conditions, and $\max(v_{ij}) - v_{ij}$ denotes adjusting the random errors of all decision units to the same state, thus removing the effects of chance factors.

In the third stage, the adjusted input–output variables are analyzed to measure the real efficiency value. The input variables of the second stage are adjusted to obtain value and initial value output variables in the BCC-DEA model, and to recalculate the financing efficiency. In this calculation, the enterprise external financing environment and the influence of random error are eliminated. This result is more objective and reliable, and closer to the real business activities in the real level of the enterprise.

## 3.2. Green Financing Efficiency Evaluation Index Selection

Determining input and output indicators and external influencing factors is the key to studying the efficiency of green financing of listed companies in the construction industry by applying the three-stage DEA method. Among these factors, the input indicators in the existing literature mainly include the scale of green financing, the structure of green financing, and the use of green financing funds, and the output indicators mainly include economic output and environmental output. However, most studies only use a single indicator to measure a certain indicator element, which is difficult to portray comprehensively. Therefore, this paper refers to Liu Dengguo and Guo Jingru (2016) [19], Wang Yao and Wang Wenwei (2019) [20], and Yang Sha et al. (2022) [21] to select secondary indicators and process them using the entropy weight method to obtain more comprehensive first-level input–output indicators.

### 3.2.1. Input Indicators

In this paper, green financing scale, green financing structure, and green financing fund use are selected as input indicators. Among them, the scale of green financing is mainly composed of green credit input and green bond input, which are mainly measured from the perspective of green financial resources being invested in the main body (listed companies in the construction industry) (Lili Zhang et al., 2018) [8]; the structure of green financing is mainly composed of total gearing ratio, short-term debt ratio, and cash current debt ratio; and the use of green financing funds is mainly composed of main operating costs and other operating costs composition.

### 3.2.2. Output Indicators

In this paper, economic output and environmental output are selected as output indicators. Among them, economic output mainly consists of return on net assets, total asset turnover ratio, and growth rate of main business income; environmental output mainly consists of carbon emission, wastewater emission, exhaust gas emission, and solid waste emission. It should be noted that the environmental output data of listed companies in the construction industry cannot be directly obtained because there is no mandatory requirement for listed companies in the construction industry to disclose carbon dioxide and waste emissions at present. Referring to the practice of scholars, the carbon dioxide emissions of listed companies in the construction industry are calculated as the ratio of the output value of listed companies in the construction industry to the output value of the construction industry, and combined with the data of carbon emissions in the construction industry in the CEADs database (Fu Hua et al., 2021) [22]. The waste emissions of listed companies in the construction industry are discounted as the proportion of the output value of listed companies in the construction industry relative to GDP (Liu Jia and Song Qiuyue, 2018; Zhang Tao and Wu Jinshang, 2021) [23,24].

### 3.2.3. External Influencing Factors

Regarding the external influencing factors of green financing efficiency, scholars have generally studied the aspects of the macroeconomic environment, financial environment, and government–enterprise relationship (Du Jinmin et al., 2016; Lu Jing et al., 2021) [25,26]. Considering that the interaction between the Chinese government and the financial market affects the green financing efficiency of enterprises, this paper introduces the interaction between the government and the financial market as an external factor influencing the green financing efficiency of listed construction companies, in addition to the above external factors. Among these factors, the macroeconomic environment is measured by the annual GDP growth rate, the financial environment is measured by the financial marketization index (Fan Gang et al., 2021) [27], the government–enterprise relationship is measured by government subsidies in non-operating income details, and the interaction between the government and the financial market is measured by the ratio of local government debt balance to regional GDP.

In summary, based on the three-stage DEA method, the selected input–output index system of green financing efficiency for listed companies in the construction industry is shown in Table 1, and the external influencing factors are shown in Table 2.

**Table 1.** Input and output indicators.

| Variables | Tier 1 Indicators | Secondary Indicators | Indicator Definition |
|---|---|---|---|
| Inputs | Green Financing Scale | Green Credit Input<br>Green Bond Input | The sum of long- and short-term bank loans<br>Sum of long- and short-term bond amounts |
| | Green Financing Structure | Total Gearing Ratio<br>Short-term Debt Ratio<br>Cash Current Liability Ratio | Ratio of total liabilities to total assets<br>Ratio of current liabilities to total assets<br>Ratio of net cash flow from operations to current liabilities |
| | Use of Green Financing Funds | Total Operating Costs<br>Cost of Main Operations | Total operating costs<br>Cost of main operations |
| Outputs | Economic Output | Return on Net Assets<br>Total Assets Turnover Ratio<br>Growth Rate of Main<br>Business Revenue | Ratio of net income to average shareholders' equity<br>Ratio of revenue from main business to total assets<br>Ratio of growth in revenue from main business to<br>revenue from main business in the previous year |
| | Environmental Output | Carbon Emissions<br>Wastewater Discharge<br>Exhaust Emissions<br>Solid Waste Emissions | Carbon dioxide emissions and normalization<br>Wastewater discharge and normalization of treatment<br>Emission of exhaust gases and normalization of treatment<br>Solid waste emissions and orthotropic treatment |

**Table 2.** External influences.

| External Influencing Factors | Indicator Definition |
|---|---|
| Macroeconomic Environment | GDP annual growth rate |
| Financial Environment | Financial Marketization Index |
| Government-Enterprise Relations | Government grants in the breakdown of<br>non-operating income |
| The Interaction between Government and<br>Financial Markets | Local government debt balance as a percentage<br>of regional GDP |

### 3.3. Sample Selection and Data Sources

This study selected the interim panel data of listed companies in the construction industry in Shanghai and Shenzhen from 2017 to 2020, and screened them according to the following criteria: (1) eliminating samples that do not meet the requirements of the green industry; (2) eliminating samples with missing key data and the existence of the ST logo in that year; (3) eliminating samples whose main business is not in the scope of the construction industry. Table 3 shows the final sample of 67 listed companies in the construction industry. The main financial indicators were obtained from the financial reports of each listed company through the CSMAR database, Wind database, Juchao Consulting website, etc. Non-financial indicators were mainly obtained from China Statistical Yearbook, China Industrial Statistical Yearbook, China Carbon Accounting Database, and China Marketization Index Database.

**Table 3.** Sample company information table.

| Serial Number | Stock Code | Serial Number | Stock Code | Serial Number | Stock Code |
|---|---|---|---|---|---|
| 1 | Northern International<br>(000065) | 24 | Sinosteel International<br>(000928) | 47 | Hongtao Corporation<br>(002325) |
| 2 | Southeast Net Frame<br>(002135) | 25 | CIGI (002051) | 48 | Yaxia Corporation<br>(002375) |
| 3 | Donghua Technology<br>(002140) | 26 | Guangdong<br>Hydropower (002060) | 49 | Guangtian Group<br>(002482) |

**Table 3.** *Cont.*

| Serial Number | Stock Code | Serial Number | Stock Code | Serial Number | Stock Code |
|---|---|---|---|---|---|
| 4 | Yanhua Intelligence (002178) | 27 | Zhejiang Jiaoke (002061) | 50 | Ruihe shares (002620) |
| 5 | Honglu Steel Structure (002541) | 28 | Hongrun Construction (002062) | 51 | Chisin Corporation (002781) |
| 6 | Sinochem Geotechnical (002542) | 29 | Chengdu Road & Bridge (002628) | 52 | China Decoration Construction (002822) |
| 7 | JiaYu stock (300117) | 30 | Pudong Construction (600284) | 53 | Meizhi Corporation (002856) |
| 8 | Haibo Heavy Science (300517) | 31 | Tibetan Skyway (600326) | 54 | Qidian Design (300500) |
| 9 | Hangxiao Steel Structure (600477) | 32 | Tengda Construction (600512) | 55 | Weiye (300621) |
| 10 | Jinggong Steel Structure (600496) | 33 | China Railway Construction (601186) | 56 | Jain Design (300668) |
| 11 | China Railway Industry (600528) | 34 | China Nuclear Construction (601611) | 57 | Jianghe Group (601886) |
| 12 | China Chemical (601117) | 35 | China CMT (601618) | 58 | Quanzhu Stock (603030) |
| 13 | Huadian Heavy Industry (601226) | 36 | China Electric Construction (601669) | 59 | Yuancheng Stock (603388) |
| 14 | Baili Technology (603959) | 37 | China Communications Construction (601800) | 60 | Collyer (603828) |
| 15 | Oriental Garden (002310) | 38 | Tianjian Group (000090) | 61 | CSC (002883) |
| 16 | Palm shares (002431) | 39 | High-tech Development (000628) | 62 | Sujiaoke (300284) |
| 17 | Pupang Stock (002663) | 40 | Shanghai Construction Engineering (600170) | 63 | CKI (300675) |
| 18 | Lingnan Corporation (002717) | 41 | Longyuan Construction (600491) | 64 | Huajian Group (600629) |
| 19 | Meichen Ecology (300237) | 42 | Chongqing Construction Industry (600939) | 65 | Tongji Technology (600846) |
| 20 | Mengcao Ecology (300355) | 43 | China Construction (601668) | 66 | Kangshe (603458) |
| 21 | Chengbang (603316) | 44 | Ningbo Construction (601789) | 67 | Hop Shing (603909) |
| 22 | Qianjing Garden (603778) | 45 | Baoying shares (002047) | | |
| 23 | Shandong Road and Bridge (000498) | 46 | Golden Mantis (002081) | | |

According to CITIC Securities Industry Classification Standard Version 2.0, 1–44 in the sample belong to the building construction industry; 45–60 belong to the construction decoration industry; 61–67 belong to the architectural design and service industry.

## 4. Measurement of Green Financing Efficiency of Listed Companies in the Construction Industry

*4.1. Initial Green Financing Efficiency Measurement in the First Stage of DEA*

Using DEAP2.1 software, the comprehensive technical efficiency, pure technical efficiency, and scale efficiency of green financing efficiency of listed companies in the con-

struction industry from 2017 to 2020 were measured using the traditional DEA method, and the results are shown in Table 4. The comprehensive technical efficiency, pure technical efficiency and scale efficiency of the initial green financing efficiency of listed companies in the construction industry in the first stage are all less than 1, although showing a fluctuating upward trend. Despite this trend, they have not reached the efficient state, i.e., they have not achieved the optimal input and output. From the decomposition factor of comprehensive technical efficiency, both pure technical efficiency and scale efficiency act on comprehensive technical efficiency; that is, the ineffective initial green financing efficiency is caused by the ineffectiveness of both pure technical efficiency and scale efficiency, and the level of listed companies in the construction industry using green financing funds needs to be improved and the scale of green financing is not reasonable enough. The average value of pure technical efficiency is smaller than scale efficiency between 2017 and 2020, which indicates that the low efficiency of initial green financing is mainly caused by the low pure technical efficiency, and the problem of the low usage of green funds by listed companies in the construction industry is prominent. The first stage of measurement does not consider the interference of external influencing factors and random errors on the results, while, in reality, the existence of unfavorable external influencing factors will lead to a lower value of green financing efficiency. Thus, the influence of external influencing factors and random errors on the initial green financing efficiency should be eliminated to measure the real green financing efficiency of listed companies in the construction industry.

**Table 4.** Average of initial green financing efficiency in the first phase.

|  | 20171 | 20172 | 20181 | 20182 | 20191 | 20192 | 20201 | 20202 |
|---|---|---|---|---|---|---|---|---|
| Technical efficiency | 0.142 | 0.421 | 0.202 | 0.045 | 0.565 | 0.456 | 0.582 | 0.530 |
| Pure technical efficiency | 0.299 | 0.496 | 0.336 | 0.340 | 0.633 | 0.578 | 0.650 | 0.612 |
| Scale efficiency | 0.688 | 0.893 | 0.811 | 0.410 | 0.919 | 0.838 | 0.922 | 0.893 |

### 4.2. Phase II SFA Removes External Influences and Random Errors

Using Frontier 4.1 software, the SFA regression models were constructed with the slack in green financing scale, slack in green financing structure, and slack in green financing fund utilization calculated based on the first-stage DEA results as the explanatory variables, and the macroeconomic environment, financial environment, government–enterprise relationship, and the interaction between government and financial markets as the explanatory variables, respectively. To maintain consistency in the direction of influence, the SFA regression equation was built with panel data by drawing on the treatment of existing studies (Ying Luo et al., 2019; Poor Liu and Jun Hao, 2021) [28,29], and the results of the model are shown in Table 5. The $\gamma$ values of green financing scale slack, green financing structure slack, and green financing fund utilization slack tend to be close to 1 and the LR values are all significant at the 1% level, rejecting the hypothesis that there is no management inefficiency. Thus, it was reasonable and necessary to conduct SFA regression analysis. The impact of each external influence factor on the initial green financing efficiency of listed companies in the construction industry was further analyzed. Since the external influence factor is a regression of each input slack variable, when the regression coefficient is negative, it means that increasing the value of the external influence factor is conducive to reducing the amount of input slack, generating savings and having a positive impact on green financing efficiency; when the regression coefficient is positive, it means that increasing the external influence factor will increase the amount of input slack, leading to increased waste and having a reverse impact on green financing efficiency. It should be added that the t-value test is not significant but still has a directional effect (Li Ran and Feng Zhongzhao, 2009) [30].

**Table 5.** Second-stage SFA regression results.

| | Green Financing Scale Slack Volume | Green Financing Structure Slack Volume | Slack in the Use of Green Financing Funds |
|---|---|---|---|
| Constant term | 0.00036 *** | −0.1055 | 0.02751 *** |
| | (3.0314) | (−0.3319) | (4.3064) |
| Macroeconomic Environment | 0.0004 * | 0.3005 ** | −0.07964 * |
| | (1.9236) | (2.4743) | (−1.9459) |
| Financial Environment | −0.00003 ** | −0.0014 | −0.00025 |
| | (−2.7220) | (−0.5599) | (−0.5975) |
| Government-Enterprise Relations | −0.00001 ** | 0.0057 *** | −0.00009 |
| | (−2.5269) | (4.9788) | (−0.2494) |
| The interaction between government and financial markets | −0.00031 *** | −0.1548 | −0.20815 *** |
| | (−3.4297) | (−1.3395) | (−7.6912) |
| $\sigma^2$ | 0.03959 | 0.0509 | 0.02729 |
| $\gamma$ | 0.99999 | 0.99999 | 0.99999 |
| log likelihood | 915.3634 | 112.5450 | 656.9205 |
| LR test | 1229.5145 *** | 31.5987 *** | 679.5357 *** |

* indicates at 10% significance level, ** at 5% significance level, *** at 1% significance level; values in parentheses are the corresponding t-values.

Specific results are discussed in the following.

(1) Macroeconomic environment. The regression results between the macroeconomic environment and the scale of green financing and the slack of green financing structure are positive and significant at the significance level of 10% and 5%, respectively. The regression coefficients between the macroeconomic environment and the slack of green financing capital use are negative and significant at the significance level of 10%. This shows that the improvement in the macroeconomic environment makes it easier for listed construction companies to obtain green funds, but it also leads to the waste of funds caused by excessive green financing and adversely affects the green financing structure. However, the good macroeconomic environment is also conducive to the listed construction companies to reduce the input cost, so that the use of green capital waste situation is improved.

(2) Financial environment. The regression results of the financial environment and green financing scale, green financing structure, and green financing fund use slack are all negative, and the regression coefficient of green financing scale slack is significant at the 5% significant level. This indicates that the improvement in the financial environment makes the financial market more rational. In addition, financial institutions tend to invest funds in companies with higher productivity, while listed companies in the construction industry will make rational decisions and improve operational efficiency to obtain green financing, thus improving the efficiency of green financing.

(3) Government–enterprise relationship. The regression coefficients of the government–enterprise relationship with green financing scale slack are negative and significant at the 5% level of significance; those with green financing structure slack are positive and significant at the 1% level of significance; and those with green financing fund utilization slack are negative but do not pass the significance test. This indicates that the improvement in the government–enterprise relationship makes it easier for listed companies in the construction industry to obtain funds from government channels, thus reducing the waste of green funds in the financial market and increasing the utilization of funds. However, the reliance on the government–enterprise relationship, especially on government subsidies, exacerbates the financial risk of listed companies in the construction industry and makes their green financing structure unbalanced.

(4) Interaction between government and the financial market. The regression results of the interaction between the government and the financial market and green financing scale, green financing structure, and green financing fund use slack are all negative, and the regression coefficients of the interaction with green financing scale and green financing fund use slack are significant at the 1% significant level. This indicates the intensification of the interaction between the government and the financial market, i.e., the increase in local government debt has led to an increase in government investment in the construction of infrastructure, etc. As the beneficiary of government infrastructure projects, listed companies in the construction industry have therefore gained more business, thus enabling them to integrate more green funds and reduce costs, which has improved the efficiency of green financing.

The regression results of SFA show that the above external influences have a more profound impact on most of the inputs. Based on the regression results of SFA, the input variables were then adjusted using Equation (4) so that all decision units were adjusted to the same environmental conditions, and the adjusted input variables provided the basis for the measurement of green financing efficiency in the third stage.

### 4.3. Phase III DEA Green Financing Efficiency Measurement

In the third stage of the three-stage DEA, to obtain the green financing efficiency of listed companies in the construction industry, the adjusted inputs obtained in the second stage and the original outputs in the first stage were used as the base data, and the corresponding results obtained are shown in Table 6, and were measured again using DEAP2.1 software. In a comprehensive view, the comprehensive technical efficiency, pure technical efficiency, and scale efficiency of the green financing efficiency of listed companies in the construction industry in the third stage from 2017 to 2020 are all less than 1. The green financing efficiency after excluding external influencing factors and random error interference still did not reach the efficient state; that is, it did not achieve the optimal input and output. However, there is a fluctuating upward trend, indicating that the green financing efficiency of the construction industry has a better development trend. Further, the mean value of green efficiency in the third stage was compared with the results of the first stage, as shown in Figure 1. It is easy to find that, compared with the first stage, the green financing efficiency of listed companies in the construction industry measured in the third stage is significantly higher, which indicates that it is reasonable and necessary to use the three-stage DEA to exclude the interference of external influences and random errors. Among these factors, the pure technical efficiency is the most disturbed, and the pure technical efficiency measured in the first stage is seriously underestimated, while the scale efficiency is relatively less affected. At the same time, the mean value of pure technical efficiency in the third stage is still lower than that of scale efficiency, and after excluding the interference of external influences and random errors, the problem of the low usage of green capital by listed companies in the construction industry is still prominent.

**Table 6.** Average of green financing efficiency in the third stage.

|  | 20171 | 20172 | 20181 | 20182 | 20191 | 20192 | 20201 | 20202 |
|---|---|---|---|---|---|---|---|---|
| Technical Efficiency | 0.538 | 0.653 | 0.506 | 0.516 | 0.695 | 0.686 | 0.719 | 0.743 |
| Pure Technical Efficiency | 0.606 | 0.685 | 0.554 | 0.607 | 0.746 | 0.736 | 0.749 | 0.766 |
| Scale Efficiency | 0.906 | 0.964 | 0.934 | 0.872 | 0.941 | 0.942 | 0.960 | 0.972 |

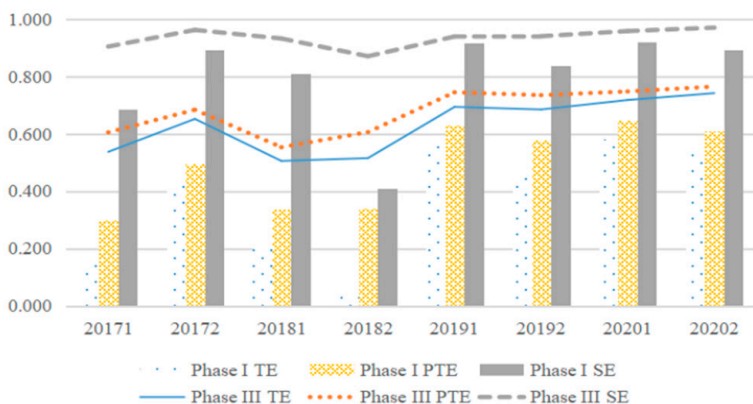

**Figure 1.** Comparison of Phase 1 and Phase 3 results for listed companies in the construction industry, 2017–2020.

## 5. Internal Influencing Factors of Green Financing Efficiency of Listed Companies in the Construction Industry

### 5.1. Internal Influencing Factor Selection

Since the measured green financing efficiency of listed companies in the construction industry is not optimal, this study attempted to further explore its internal influencing factors in order to make targeted suggestions for improvement. Therefore, the possible internal influencing factors of green financing efficiency of listed companies in the construction industry were selected, are divided into two categories: corporate characteristics and executive characteristics. These variables are defined as follows.

### 5.1.1. Enterprise Characteristics

To improve the efficiency of green financing, listed construction companies must have certain resources and capabilities, which are reflected in enterprise size, debt maturity structure, ownership concentration, R&D, and innovation ability. Enterprise scale is measured by the natural logarithm of main business income. The debt maturity structure is measured by the ratio of current liabilities to total liabilities. Ownership concentration is measured by the shareholding ratio of the largest shareholder. R&D and innovation capacity are measured by the ratio of R&D investment to main business income.

### 5.1.2. Executive Characteristics

Based on the upper echelon theory, managers' characteristics influence their strategic choices and then influence the behavior of firms (Su Weihua et al., 2022) [31]. Therefore, the characteristics of senior executives will have an impact on the green financing efficiency of listed construction companies. In this study, the government background and financial background of senior executives were taken as the focus of the research, and dummy variables were used that take a value of 1 for yes, and 0 for no. Indicators and descriptive analyses of internal influencing factors of green financing efficiency are shown in Table 7.

**Table 7.** Internal influencing factors and descriptive statistics.

| Variables | Symbols | Number of Samples | Average Value | Standard Deviation | Minimum Value | Maximum Value |
|---|---|---|---|---|---|---|
| Enterprise size | Size | 536 | 23.27 | 1.717 | 19.86 | 28.42 |
| Debt maturity structure | LLD | 536 | 0.537 | 0.135 | 0.099 | 0.832 |
| Ownership concentration | Topic1 | 536 | 0.366 | 0.149 | 0.0826 | 0.765 |
| R&D and innovation capability | R & D | 536 | 0.035 | 0.026 | 0 | 0.205 |
| corporate executives with government background | Polcon | 536 | 0.179 | 0.384 | 0 | 1 |
| corporate executives with financial background | Finback | 536 | 0.127 | 0.333 | 0 | 1 |

### 5.2. Empirical Model

Combining the green financing efficiency of listed companies in the construction industry measured above and the selected internal influencing factors, including enterprise size, debt maturity structure, ownership concentration, R&D, and innovation capability, corporate executives with a government background, and corporate executives with a financial background, the dynamic panel model of this study was set as follows:

$$TE_{it} = \alpha_0 + \alpha_1 TE_{i,t-1} + \alpha_2 Size_{it} + \alpha_3 LLD_{it} + \alpha_4 Topic1_{it} + \alpha_5 R\&D_{it} + \alpha_6 Polcon_{it} + \alpha_7 Finback_{it} + u_i + \lambda_t + \varepsilon_{it} \quad (5)$$

where $TE_{it}$ is the green financing efficiency value of listed companies in the construction industry, $TE_{i,t-1}$ is the lagged green financing efficiency value, $\alpha_0$ is a constant term, $\alpha_1$ to $\alpha_6$ are coefficients to be estimated, $u_i$ denotes the unobservable individual effect, $\lambda_t$ denotes the time effect, and $\varepsilon_{it}$ is the random error term.

The model with the introduction of the lagged terms of the explanatory variables has dynamic explanatory power. However, there is also an endogeneity problem in the model, and the estimation results will be biased and inconsistent if the traditional methods are used for estimation. In this study, the systematic GMM method, which can effectively mitigate the endogeneity problem caused by the first-order lagged terms of the explanatory variables and the correlation of panel effects, was selected for estimation. Moreover, considering that the two-step systematic GMM method is more effective in dissipating the endogeneity problem of the model compared with the one-step method, this study used the two-step systematic GMM method to test the estimation of the model.

### 5.3. Empirical Results and Analysis

This study used the systematic GMM two-step method to analyze the internal influencing factors of green financing efficiency of listed companies in the construction industry, and the results are shown in Table 8. The *p*-value of AR (1) is less than 0.1 and the *p*-value of AR (2) is greater than 0.1, which indicates that there is no second-order serial autocorrelation in the error term, and thus the model can effectively overcome the endogeneity problem. The over-identification constraint test was performed using the explanatory variable lagged by one period as the instrumental variable, and the *p*-value of the Hansen test result is 0.334, which indicates that the instrumental variable of the model is valid, and there is no over-identification problem. The lagged period coefficients of the explanatory variables are significantly positive, confirming that the green financing efficiency of listed companies in the construction industry has a strong inertia and is influenced not only by the relevant factors in the current year, but also by the green financing efficiency in the previous period. All the selected explanatory variables pass the significance test, except for the corporate executives with a financial background, among which, ownership concentration

and corporate executives with a government background have positive effects on the green financing efficiency of listed companies in the construction industry, and enterprise size, debt maturity structure, R&D, and innovation capability have negative effects.

**Table 8.** Systematic GMM regression results and robustness tests.

| Indicators | Systematic GMM Regression Results (1) | Robustness Test 1 | Robustness Test 2 |
|---|---|---|---|
| TE (−1) | 0.3484099 *** | 0.2761024 * | 0.3494548 ** |
| | (3.53) | (2.63) | (3.44) |
| Size | −0.0782126 * | −0.0850384 * | −0.0904901 ** |
| | (−2.54) | (−2.09) | (−2.90) |
| LLD | −1.131748 *** | −1.082979 *** | −1.089512 *** |
| | (−4.40) | (−3.80) | (−3.92) |
| Top1 | 0.5240953 ** | 0.4156255 * | 0.6104463 ** |
| | (3.03) | (2.26) | (3.32) |
| R & D | −2.827662 *** | −2.611893 ** | −2.76041 *** |
| | (−3.81) | (−2.86) | (−3.76) |
| Polcon | 0.1976802 ** | 0.2088119 * | 0.1910611 ** |
| | (3.00) | (2.60) | (2.75) |
| Finback | 0.1160204 | 0.1537654 | 0.1242112 |
| | (1.88) | (1.89) | (1.77) |
| TAT | - | 0.3974024 ** | - |
| | | (2.87) | |
| AR (1) *p*-value | 0.000 | 0.000 | 0.000 |
| AR (2) *p*-value | 0.156 | 0.824 | 0.137 |
| Hansen test *p*-value | 0.629 | 0.328 | 0.639 |

* indicates at 10% significance level, ** at 5% significance level, *** at 1% significance level; values in parentheses are the corresponding t-values.

Further analysis of the overall regression results of the system GMM was carried out.

First, enterprise size inversely affects green financing efficiency, and it is significant at the 10% level. As mentioned above, the scale efficiency of listed companies in the construction industry is already at a relatively high level; in addition, due to the problems of transformation burden of former product lines and internal organizational rigidity of large-scale listed companies in the construction industry, which makes it more difficult for them to carry out green transformation, enterprise size inversely affects the green financing efficiency.

Second, debt maturity structure inversely affects green financing efficiency, and it is significant at the 1% level. When the proportion of current liabilities is high, listed companies in the construction industry may have the behavior of "short money and long use" and irrational investment, which reduces the green financing efficiency.

Third, ownership concentration positively affects green financing efficiency, and it is significant at the 5% level. For listed companies in the construction industry at the current stage, increasing ownership concentration can reduce the phenomenon of "free-riding" of small and medium shareholders and prevent management from self-interest behaviors that are harmful to shareholders' interests. In addition, major shareholders can influence the board of directors to select more capable managers and actively participate in the company's business decisions, which has a positive impact on the green financing efficiency.

Fourth, R&D and innovation capability inversely affect green financing efficiency and is significant at the 1% level. The innovation capacity resulting from R&D investment has uncertainty and lag, and the current R&D and innovation capacity does not reach the level of promoting green financing efficiency due to the threshold effect.

Fifth, corporate executives with a government background positively affect green financing efficiency, and this variable is significant at the 5% level. When the executives of listed companies in the construction industry have a government background, their political sensitivity and "political connections" will encourage them to grasp the direction of industrial policies and promote the green development of the company "in line with

the trend". At the same time, their social influence helps companies to obtain better green financing and thus improve the green financing efficiency.

*5.4. Robustness Tests*

In this study, the following two methods were used together for robustness testing.

First, supplementary variables were used. The total asset turnover ratio was added as a supplementary variable, denoted by the symbol TAT, and the original variables were kept unchanged and the regression was re-run using the two-step systematic GMM method.

Second, the special sample was excluded. The two sample companies with the highest and lowest green financing efficiency were excluded. The original variables were left unchanged and the regression was re-run using the two-step systematic GMM method.

As shown in Table 8, the direction and significance of the regression coefficients are generally consistent with the results of the systematic GMM regression (1), which passed the robustness test.

## 6. Expanded Analysis

The above results confirmed that the green financing efficiency of listed companies in the construction industry is not optimal, and the related internal and external influencing factors have been discussed. In the extended analysis, this study furthered explore two aspects: First, what is the level of green financing efficiency of listed companies in the construction industry by industry segment? Second, what factors affect the green financing efficiency of listed companies in this sector?

*6.1. Green Financing Efficiency by Industry Segment*

As the division of labor in society continues to evolve, the construction industry, as a large-scale industry, is being refined within its various subsectors. To further explore the industry differences in green financing efficiency and the influencing factors of listed companies in the construction industry in the context of carbon neutrality, this section presents the study of the subdivided industries in the construction industry. According to the CITIC Securities industry classification rules and the differences in carbon emissions, the construction industry is subdivided into three categories: the building construction industry, the construction decoration industry, and the architectural design and service industry. Furthermore, the green financing efficiency of the subdivided industries was measured according to the above empirical steps; due to the limitation of space, only the third-stage results are shown in this section. As can be seen from Table 9, the green financing efficiency of listed companies in the three subsectors is roughly between 0.6 and 1, with relatively large fluctuations, and all listed companies in the three subsectors have not achieved green financing efficiency, especially the lowest with pure technical efficiency. By 2020, using technical efficiency as the evaluation standard, listed companies in the architectural design and service industry had the highest green financing efficiency, listed companies in the construction decoration industry had the second-highest, and the efficiency of the listed companies in the building construction industry was relatively low. Among these companies, listed companies in the building construction industry and construction decoration industry are mainly restricted by pure technical efficiency, and the level of green capital application is low; listed companies in the architectural design and service industry are mainly restricted by scale efficiency, and the scale of green financing needs to be improved.

**Table 9.** Average of green financing efficiency in the third stage by subsector.

| Segmentation | | 20171 | 20172 | 20181 | 20182 | 20191 | 20192 | 20201 | 20202 |
|---|---|---|---|---|---|---|---|---|---|
| Building Construction Industry | Integrated technical efficiency | 0.739 | 0.700 | 0.701 | 0.767 | 0.841 | 0.843 | 0.731 | 0.755 |
| | Pure technical efficiency | 0.771 | 0.726 | 0.750 | 0.792 | 0.857 | 0.854 | 0.771 | 0.774 |
| | Scale efficiency | 0.955 | 0.967 | 0.941 | 0.970 | 0.981 | 0.987 | 0.945 | 0.976 |
| Construction Decoration Industry | Integrated technical efficiency | 0.667 | 0.669 | 0.612 | 0.573 | 0.774 | 0.711 | 0.795 | 0.807 |
| | Pure technical efficiency | 0.741 | 0.774 | 0.683 | 0.722 | 0.845 | 0.817 | 0.847 | 0.842 |
| | Scale efficiency | 0.922 | 0.888 | 0.928 | 0.839 | 0.926 | 0.891 | 0.946 | 0.963 |
| Architectural Design and Services Industry | Integrated technical efficiency | 0.717 | 0.746 | 0.794 | 0.829 | 0.783 | 0.844 | 0.830 | 0.846 |
| | Pure technical efficiency | 0.913 | 0.910 | 0.933 | 0.935 | 0.890 | 1.000 | 0.987 | 0.930 |
| | Scale efficiency | 0.804 | 0.836 | 0.861 | 0.893 | 0.892 | 0.844 | 0.841 | 0.916 |

*6.2. Factors Influencing the Efficiency of Green Financing by Sector*

6.2.1. External Influencing Factors

In the analysis of external influence factors of listed companies in the construction industry segment, the results of the second-stage SFA regression in the three-stage DEA analysis were used as the results of external influence factors by referring to Yu, Hongwei and Hu, Dezhu (2015) [32], and Guo, Si-Dai et al. (2018) [33]. Due to space limitation, this section only shows the regression results for listed companies in the architectural design and service industry, as shown in Table 10.

**Table 10.** SFA regression results for listed companies in the architectural design and services industry.

| Relaxation Variables | Green Financing Scale Slack Volume | Green Financing Structure Slack Volume | Slack in the Use of Green Financing Funds |
|---|---|---|---|
| Constant term | 0.0046 (0.0046) | −0.1025 ** (−2.4165) | 0.0042 (0.0042) |
| Macroeconomic Environment | −0.0013 (−0.0013) | −0.0457 (−0.7722) | −0.0003 (−0.0003) |
| Financial Environment | −0.0002 (−0.0002) | 0.0038 ** (2.3297) | −0.0002 (−0.0002) |
| Government–Enterprise Relations | 0.0000 (0.0000) | 0.0002 (0.3248) | 0.0000 (0.0000) |
| The interaction between government and financial markets | −0.004 (−0.004) | 0.1053 *** (2.7565) | −0.0037 (−0.0037) |
| $\sigma^2$ | 0.0000 | 0.1504 | 0.0000 |
| $\gamma$ | 0.9500 | 0.99999 | 0.9300 |
| log likelihood | 303.5360 | 49.8943 | 321.4315 |
| LR test | 19.7670 *** | 71.4310 *** | 8.9542 *** |

** at 5% significance level, *** at 1% significance level; values in parentheses are the corresponding t-values.

The external influences on listed companies are discussed in this section.

Listed companies in the building construction industry. First, the improvement in the macroeconomic environment helps them to make scientific decisions and reasonable financing, so as to reduce the waste of green funds and lower the input cost. Second, the improvement in the financial environment helps them to make more rational decisions, so that they can control the scale of green financing. However, their use of green financing

funds is problematic and there are wasteful uses of resources. Third, the intensification of the interaction between the government and the financial market, i.e., the elevation of local government debt, has led to an increase in infrastructure projects, which, to some extent, helps them to obtain more business and improves the green financing efficiency.

Listed companies in the construction decoration industry. The impact of the external environment on listed companies in the construction and renovation industry is not significant and will not be analyzed.

Listed companies in the architectural design and service industry. First, when the financial environment is favorable, they may be over-optimistic and have a higher percentage of debt, thus leading to an imbalanced green financing structure and risk. Second, the increasing interaction between the government and the financial market not only increased their resources, but also squeezed their financial resources, making their financing more difficult and expensive, and increasing the risk of the green financing structure.

6.2.2. Internal Influencing Factors

In the analysis of internal influencing factors of green financing efficiency of listed companies by subsector, the empirical steps and methods for listed companies in the building construction industry and construction decoration industry were the same as the overall sample of the construction industry. In contrast, the data set used for listed companies in the architectural design and service industry is a long panel, and the use of the systematic GMM method may make the results more biased; therefore, the bias-corrected LSDV method (Rui Huang et al., 2021) [34] was used for the analysis, and the results are shown in Table 11. Due to space limitations, the descriptive statistics and model tests in this section are not repeated here.

**Table 11.** Regression results of internal influencing factors of subsectors.

| Indicators | Building Construction Industry | Construction Decoration Industry | Architectural Design and Services |
|---|---|---|---|
| TE (−1) | 0.4370084 *** (4.18) | 0.0335 (0.29) | 0.2851899 (1.78) |
| Size | −0.0283739 *** (−3.90) | 0.1031867 (1.93) | 0.1970367 * (2.54) |
| LLD | −0.3105997 ** (−2.87) | −2.318131 ** (−3.66) | −0.0425649 (−0.09) |
| Top1 | 0.0756907 (0.79) | 1.574112 1.32 | −4.705155 (−1.01) |
| R & D | −1.125511 * (−2.60) | −3.245929 (−1.96) | 1.276138 (1.17) |
| Polcon | 0.1081055 * (2.33) | 0.024814 (0.28) | - |
| Finback | 0.0063604 (0.14) | −0.181866 (−1.27) | - |
| AR (1) *p*-value | 0.000 | 0.024 | - |
| AR (2) *p*-value | 0.205 | 0.429 | - |
| Hansen test *p*-value | 0.574 | 0.996 | - |

* indicates at 10% significance level, ** at 5% significance level, *** at 1% significance level; values in parentheses are the corresponding t-values.

(1) Listed companies in the building construction industry. First, enterprise size has a significant reverse effect on the green financing efficiency, and, at present, these companies should not continue to expand their scale. Second, the debt maturity structure has a significant reverse effect on the green financing efficiency, and these companies should remove their reliance on short-term loans, reduce the proportion of short-term liabilities, and try to obtain long-term loans as green financing channels. Third, the R&D and innovation ability has a significant reverse effect on the green financing efficiency, so these companies should not blindly invest in R&D funds, and should further evaluate the R&D projects that take a long time and are characterized

by a slow transformation. Fourth, corporate executives with government background have a significant and positive effect in the green financing efficiency, and these companies can consider including people with government work experience in the executive team to improve the company's social influence and sensitivity to policies, and thus improve the green financing efficiency.

(2) Listed companies in the construction decoration industry. The debt maturity structure has a significant reverse effect on the green financing efficiency, and these companies should remove their reliance on short-term loans and expand the scale of long-term green financing funds, so as to reduce the risk of the debt maturity structure and improve the green financing efficiency.

(3) Listed companies in the architectural design and service industry. Enterprise size significantly and positively affects the green financing efficiency, which means these companies should increase their scale of green financing.

## 7. Conclusions and Recommendations

This study combined the three-stage DEA method and the systematic GMM method to analyze the green financing efficiency and influencing factors of 67 listed companies in the construction industry in China from 2017 to 2020. Furthermore, the green financing efficiency and influencing factors of listed companies in construction industry segments were further explored, and the following analysis results were obtained.

(1) The overall green financing efficiency of the construction industry in 2017–2020 showed a fluctuating upward trend, but did not reach the efficient state, and the key to its improvement lies in the pure technical efficiency.

(2) There are obvious differences in green financing efficiency among subsectors, and using technical efficiency as the evaluation criterion, listed companies in the architectural design and service industry showed relatively high green financing efficiency, followed by the construction decoration industry, and the building construction industry was ranked last. Listed companies in the building construction and construction decoration industry are mainly constrained by pure technical efficiency, and listed companies in the architectural design and service industry are mainly constrained by scale efficiency.

(3) Among the external influencing factors, for listed companies in the construction industry, the financial environment and the interaction between the government and the financial market have a significant positive impact on the green financing efficiency, whereas the macroeconomic environment and the relationship between the government and enterprises have a complex impact on the green financing efficiency. Among the subsectors, for listed companies in the building construction industry, the macroeconomic environment, financial environment, and the interaction between the government and the financial market have a significant positive impact on green financing efficiency; for listed companies in the construction decoration industry, the external influences are not significant; for listed companies in the architectural design and service industry, the financial environment and the interaction between the government and the financial market have a significant and negative impact on green financing efficiency.

(4) Among the internal influencing factors, for listed companies in the construction industry, ownership concentration and corporate executives with government background have a significant positive influence on green financing efficiency, whereas enterprise size, debt maturity structure, R&D, and innovation capability have a significant negative influence. Among the subsectors, for listed companies in the building construction industry, corporate executives with government background have a significant positive effect on green financing efficiency, whereas enterprise size, debt maturity structure, R&D, and innovation ability have a significant negative effect; for listed companies in the construction decoration industry, debt maturity structure has a significant negative effect on their green financing efficiency; for listed companies in

the architectural design and service industry, enterprise size has a significant positive effect on their green financing efficiency.

In the context of carbon neutrality, the green development of the construction industry is an important national industrial strategy. In addition, the construction industry is at an important juncture of urgent green transformation, at which time the improvement in green financing efficiency is of great significance. Based on this, combined with the above research findings, this paper proposes relevant suggestions for government departments and construction enterprises.

For government departments: In the situation of a macroeconomic downturn and high local government debt, financial support for green financing in the construction industry should first be increased, especially in the framework of green finance to improve its green financing environment. Second, the relationship between the government and enterprises should be handled scientifically: the construction industry should be given appropriate subsidies for green development, while policy dependency should be prevented.

For construction companies: First, they should finance within their capacity and should not blindly expand their scale. Second, they should pay attention to financial risks, change the short-term debt maturity structure, and reduce their reliance on short-term loans. Third, they should maintain the concentration of equity and enhance the shareholders' monitoring enthusiasm to improve the rationality and effectiveness of corporate decision making. Fourth, they should emphasize the synergistic management of green innovation inputs and outputs, improve the transformation of R&D results, and stop time-consuming and low-output projects. Fifth, talents with a government background should be included in the senior management team, and the corporate executives should also improve their sensitivity to policies.

**Author Contributions:** Conceptualization, Y.Y. (Yaguai Yu) and Y.Y. (Yina Yan); methodology, P.S.; validation, Y.Y. (Yina Yan); formal analysis, Y.Y. (Yina Yan); resources, Y.L.; data curation, T.N.; writing—original draft preparation, Y.Y. (Yina Yan) and P.S.; writing—review and editing, Y.Y. (Yaguai Yu), Y.L. and T.N.; visualization, Y.L.; supervision, T.N.; project administration, Y.L.; funding acquisition, Y.Y. (Yaguai Yu). All authors have read and agreed to the published version of the manuscript.

**Funding:** Ningbo Soft Science Project "Research on Mechanism and Path of Green Innovation in Manufacturing Industry under Double Carbon Strategy" (2022R020). Ningbo Philosophical and Social Science Planning "Research on Green Innovation Path of Ningbo Specialized Small Giant Enterprises under Double Carbon Strategy" (G2022-2-70). Zhejiang Philosophy and Social Science Planning Project "Research on Synergistic Theory and Policy Innovation of Pollution Reduction and Carbon Emission" (22NDYDO41YB). Zhejiang Philosophy and Social Science Planning Project "Research on construction mechanism and path of Ecological Civilization Highland in Zhejiang". Longyuan Construction Finance Institute of Ningbo University Project "Research on Impact of Internet Finance Development on Development of Construction Industry" (LYZDA2002).

**Institutional Review Board Statement:** Not applicable.

**Informed Consent Statement:** Not applicable.

**Data Availability Statement:** Data is contained within the article.

**Conflicts of Interest:** The authors declare no conflict of interest.

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
