# Peer review of "Green Financing Efficiency and Influencing Factors of Chinese Listed Construction Companies against the Background of Carbon Neutralization: A Study Based on Three-Stage DEA and System GMM"

_axioms, doi:10.3390/axioms11090467_

Round 1

Reviewer 1 Report

Some comments are as follows:

1)It is suggested to add a paragraph to illustrate the innovation of this paper.

2)What is the underlying idea for the BCC-DEA model?

Author Response

Dear reviewer:

Re: Manuscript ID: axioms-1870174 and Title: Green Financing Efficiency and Influencing Factors of Chinese Listed Construction Companies under the Background of Carbon Neutralization:A Study Based on Three-Stage DEA and System GMM

Thank you for your comments concerning our manuscript entitled “Green Financing Efficiency and Influencing Factors of Chinese Listed Construction Companies under the Background of Carbon Neutralization:A Study Based on Three-Stage DEA and System GMM” (axioms-1870174). Those comments are valuable and very helpful. We have read through comments carefully and have made corrections. Based on the instructions provided in your letter, we uploaded the file of the revised manuscript. Revisions in the text are shown using red highlight for additions. The responses to the reviewer's comments are marked in red and presented following.

Point 1: It is suggested to add a paragraph to illustrate the innovation of this paper.

Response 1: We are grateful for the suggestion. To be more clearly and in accordance with the reviewer concerns, we have added a more detailed interpretation regarding the innovation of this paper in Part II.

Point 2: What is the underlying idea for the BCC-DEA model?

Response 2: Thank you very much for your advice. We added a more detailed introduction about the BCC-DEA Model in 3.1 of the article.

We would love to thank you for allowing us to resubmit a revised copy of the manuscript and we highly appreciate your time and consideration.

Sincerely.

Reviewer 2 Report

This paper combined the green industrial strategy and green financial policies for the construction industry implemented in China in the context of carbon neutrality, takes 67 listed companies in the construction industry from 2017 to 2020 as a research sample, measures the green financing efficiency and identifies its influencing factors based on the three-stage DEA and systematic 15 GMM method.

The idea of the paper is unique and well addressed. There sis no minor suggestions. I am happy to accept this paper as it is.

Author Response

Dear reviewer:

Re: Manuscript ID: axioms-1870174 and Title: Green Financing Efficiency and Influencing Factors of Chinese Listed Construction Companies under the Background of Carbon Neutralization:A Study Based on Three-Stage DEA and System GMM

Thank you for your comments concerning our manuscript entitled “Green Financing Efficiency and Influencing Factors of Chinese Listed Construction Companies under the Background of Carbon Neutralization:A Study Based on Three-Stage DEA and System GMM” (axioms-1870174). Those comments are valuable and very helpful. We have read through comments carefully and have made corrections. Thank you very much for your recognition of our article.

We would love to thank you for allowing us to resubmit a revised copy of the manuscript and we highly appreciate your time and consideration.

Sincerely.

Reviewer 3 Report

Keep the good work, interesting study. I'd recommend deepening the study in other areas beside the construction area, and for another economies, to confirm if the results are like the ones find in China.

Author Response

Dear reviewer:

Re: Manuscript ID: axioms-1870174 and Title: Green Financing Efficiency and Influencing Factors of Chinese Listed Construction Companies under the Background of Carbon Neutralization:A Study Based on Three-Stage DEA and System GMM

Thank you for your comments concerning our manuscript entitled “Green Financing Efficiency and Influencing Factors of Chinese Listed Construction Companies under the Background of Carbon Neutralization:A Study Based on Three-Stage DEA and System GMM” (axioms-1870174). Those comments are valuable and very helpful. We have read through comments carefully and have made corrections. Based on the instructions provided in your letter, we uploaded the file of the revised manuscript. Revisions in the text are shown using red highlight for additions. The responses to the reviewer's comments are marked in red and presented following.

Point 1: Keep the good work, interesting study. I'd recommend deepening the study in other areas beside the construction area, and for another economies, to confirm if the results are like the ones find in China.

Response 1: We are grateful for the suggestion. We will continue to conduct relevant research in other industries and regions.

We would love to thank you for allowing us to resubmit a revised copy of the manuscript and we highly appreciate your time and consideration.

Sincerely.
